# Hydrangea paniculata coumarins attenuate experimental membranous nephritis by bidirectional interactions with the gut microbiota

Zhaojun Li[1,2,4], Xingguang Zhang[3,4], Haijie Wu[1,4], Zhiling Ma[1], Xikun Liu[1], Jie Ma[1], Dongming Zhang[1], Li Sheng [1,5✉], Xiaoguang Chen [1,5✉] & Sen Zhang [1,5✉]

Coumarins isolated from *Hydrangea paniculata* (HP) had a renal protective effect in experimental membranous nephritis (MN), but the mechanisms are not clear. Currently, we investigate whether the modulation of gut dysbiosis by HP contributes to its renal protection. Experimental MN rats were treated with HP for six weeks. Fecal 16S rDNA sequencing and metabolomics were performed. Fecal microbiota transplantation (FMT) was used for the evaluation study. The results demonstrate that deteriorated renal function and gut dysbiosis are found in MN rats, as manifested by a higher Firmicutes/Bacteroidetes ratio and reduced diversity and richness, but both changes were reversed by HP treatment. Reduced gut dysbiosis is correlated with improved colonic integrity and lower endotoxemia in HP-treated rats. HP normalized the abnormal level of fecal metabolites by increasing short-chain fatty acid production and hindering the production of uremic toxin precursors. FMT of HP-treated feces to MN animals moderately reduced endotoxemia and albuminuria. Moreover, major coumarins in HP were only biotransformed into more bioactive 7-hydroxycoumarin by gut microbiota, which strengthened the effect of HP in vivo. Depletion of the gut microbiota partially abolished its renal protective effect. In conclusion, the bidirectional interaction between HP and the gut microbiota contributes to its beneficial effect.

[1] State key laboratory of bioactive substances and functions of natural medicines, Institute of Materia Medica, Chinese Academy of Medical Sciences & Peking Union medical college, Beijing 100050, China. [2] Department of Medicine Solna, Center for Molecular Medicine, Karolinska University Hospital and Karolinska Institutet, Stockholm, Sweden. [3] Department of Endocrinology, The seventh medical center of Chinese PLA General Hospital, Beijing 100070, China. [4]These authors contributed equally: Zhaojun Li, Xingguang Zhang, Haijie Wu. [5]These authors jointly supervised this work: Li Sheng, Xiaoguang Chen, Sen Zhang. ✉email: shengli@imm.ac.cn; chxg@imm.ac.cn; zhangs@imm.ac.cn

As a traditional Chinese medicine and a local food additive to improve flavor, *Hydrangea paniculata* has been used in southern China to treat inflammation and kidney disease. The water extract of *H. paniculata* obtained by an optimal procedure, abbreviated as HP, is currently in preclinical development for chronic kidney disease (CKD) as a class 1.3 natural medicine. Although HP contains a mixture of more than a dozen chemical compounds, 95% of its chemical constituents have been identified, as described in our previous studies[1,2]. Briefly, coumarin derivatives account for nearly 80% of these compounds, including skimmin and apiosyskimmin, which together account for approximately 55% of the constituents. Coumarins are naturally occurring sweet-smelling compounds that exist in many food plants[3]. Recent experimental results in our laboratory demonstrated that HP has a beneficial effect in cationized-BSA (c-BSA) -induced experimental membranous nephritis (MN), mainly mediated by the inhibition of complement activation and the attenuation of macrophage-mediated fibrosis[4]. Moreover, acute and long-term toxicity using rats[2,4] and beagle dogs also confirmed that HP has limited and acceptable toxicity. Therefore, HP is expected to be successful in future drug clinical trials.

Currently, increasing evidence suggests that gut microbiota dysbiosis is associated with CKD, especially immune-mediated CKD[5]. Dysbiosis of the gut microbiota deteriorates kidney function by altering gut bacterial diversity and abundance, reducing the production of short-chain fatty acids (SCFAs), impairing intestinal barrier integrity and causing bacterial translocation, which eventually triggers a state of persistent systemic inflammation in CKD patients[5]. Harmful gut microbiota-derived metabolites, such as p-cresyl sulfate, trimethylamine-N-oxide, indoxyl sulfate and indole-3 acetic acid, also contribute to the progression of CKD and have been proven to be uremic toxins that increase CKD risk[6,7]. Therefore, modulating the gut microbiota has become an important option and adjuvant therapy strategy for slowing the progression of CKD.

Increasing evidence suggests that some medicinal plant extracts and natural compounds have bidirectional interactions with the gut microbiota. On the one hand, natural compounds can reshape and normalize the dysbiosis of the gut microbiota to exert their beneficial effect;[8,9] on the other hand, gut microbiota might bio transform natural compounds into more active metabolites, thus enhancing their bioactivities[8], and the existence of gut microbiota is necessary for the pharmacological effects. Biotransformation occurs both in terms of natural compounds and gut microbiota composition; these reciprocal effects are termed the "bidirectional interaction". From a pharmacokinetic perspective, this bidirectional interaction is a possible explanation of why natural compounds with low oral bioavailability or low bioactivities in vitro have satisfactory effects in experimental animal models in vivo[10]. Our previous pharmacokinetic study in rats and mice demonstrated that HP has relatively less bioavailability after oral administration. Thus, we hypothesized that the beneficial effect of HP in the kidney might be at least partially mediated by gut microbiota modulation. Previous studies from our laboratory have demonstrated that one major metabolite of HP, 7-hydroxycoumarin (7-HC), which is enriched in the kidney, is more bioactive than skimmin and apiosylskimmin in terms of anti-inflammation and anti-oxidation, but whether its production is dependent on gut microbiota metabolism is not clear. In the current study, using an in vitro method, we will clarify the role of the gut microbiota.

The main purpose of this study was to clarify whether oral administration of HP could reverse gut dysbiosis in c-BSA-induced MN rats and confirm that the beneficial effect of HP in the kidney is partially dependent on its modulation of the gut microbiota. The second aim was to investigate whether HP could promote intestinal immunity and integrity by reducing the metabolic dysbiosis of the gut microbiota; the third aim was to determine whether the gut microbiota bio transforms major coumarin derivates from HP into 7-hydroxylcoumarin.

## Results

### HP reduces proteinuria and ameliorates kidney pathological injuries in c-BSA induced MN.
Before drug administration, the body weight and baseline albuminuria of each animal were recorded, and the difference was not remarkable on both parameters among all the experimental MN groups (Supplementary Fig. S1). After treatment for six weeks, HP significantly reduced the ratio of urinary albumin/creatinine, as well as serum neutrophil gelatinase associated lipocalin (NGAL), blood urea nitrogen (BUN), serum creatine (Scr), total cholesterol and Kidney index dose dependently (Supplementary Fig. S2). Significant glomerular lesions were observed in the model group, such as glomerular hypertrophy with thickening of the glomerular capsule wall, and partial glomerular sclerosis (Supplementary Fig. S3). HP had an obvious effect to attenuate glomerular hypertrophy and sclerosis (Supplementary Fig. S3-S4). The tubulointerstitial damage was characterized by protein cast, lymphocyte infiltration, and tubular vacuolar degeneration, and HP remarkably reduced the tubulointerstitial damage score dose dependently, and the higher dose group of HP had the same effect as mycophenolate mofetil (MMF) at 20 mg/kg dosage (Supplementary Fig. S3—S4).

Transmission electron microscope (TEM) was used to observe the ultrastructural changes of the kidney. Compared with sham rats, the podocyte integrity in model group disappeared and the glomerular basement membrane (GBM) was thicker caused by immune complex deposit. HP had a remarkable effect to reduce GBM thickness and maintained podocyte integrity (Supplementary Figs. S3—S4). These results imply that HP has satisfactory renal protective effect on experimental MN animals.

### HP reduces systemic and renal inflammation in c-BSA induced MN rats.
Intravenous c-BSA challenge caused both systemic and renal inflammation during the MN pathogenesis. Thirteen cytokines and chemokines including lnterleukin-1α (IL-1α), IL-1β, IL-6, IL-10, IL-12p70, IL-17A, IL-18, IL-33, chemokine (C-X-C motif) ligand 1 (CXCL1), monocyte chemotactic protein-1 (MCP-1), granulocyte colony-stimulating factor (GM-CSF), interferon-γ (IFN-γ) and tumor necrosis factor receptor (TNF-α) were quantified using The LEGENDplex™ Rat Inflammation Panel. By this kit, we demonstrated that IFN-γ, CXCL-1, MCP-1, TNFα, GM-CSF, IL-18, IL17A, IL33, IL-6 and IL-1α were significantly increased in serum of MN rats compared with sham control (Fig. 1a). Meanwhile, HP treatment significantly reduced these serum cytokines and chemokines in a dose-dependent manner (Fig. 1a), revealing that HP treatment attenuated systemic inflammation. These tested cytokines represented biomarkers for different subtype inflammatory cells, such as type 1 T helper (Th1) dominant (TNFα, INF-γ and GM-CSF), type 2 T helper (Th2) dominant (IL6, IL33), macrophage dominant (IL-1α, IL-18, MCP-1 and CXCL1), and type 17 T helper (Th17) dominant IL-17A.

Infiltration of immune cells into kidneys and consequently caused local inflammation contribute to the progression of MN. The data revealed that HP had an obvious effect to reduce the renal *IFN-γ, CXCL-1, MCP-1, TNFα, GM-CSF, IL-18, IL-13, IL17A, IL-33, IL-22, IL-6 and IL-1α* using real-time polymerase chain reaction (PCR), which are important inflammatory mediators in MN pathogenesis (Fig. 1b). HP treatment decreasing renal MCP-1 was consistent with reducing CD68+ macrophage infiltration into kidneys, which was indicated by immunohistochemistry (Supplementary Fig. S5). Combined with reduced

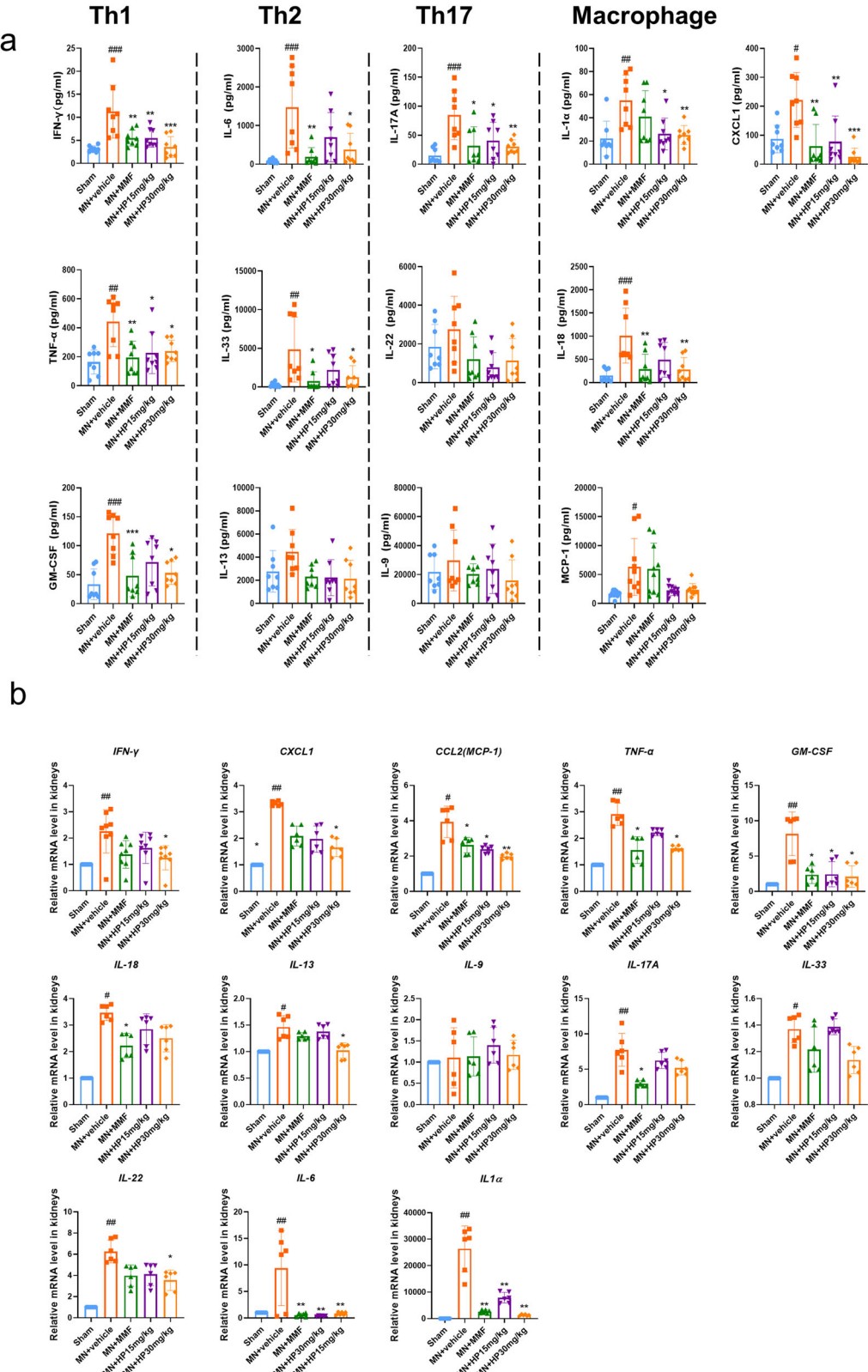

**Fig. 1 HP decreases pro-inflammatory cytokine and chemokine levels in the peripheral blood and kidney tissues in cBSA-induced MN rats. a** Protein levels of cytokines and chemokines in peripheral blood examined by LEGENDplex™ Rat Inflammation Panel, ($n = 8$). **b** mRNA levels of cytokines and chemokines in kidney tissues by qRT-PCR, ($n = 6$). #$P < 0.05$, ##$P < 0.01$, ###$P < 0.001$ versus sham group. *$P < 0.05$, **$P < 0.01$, versus vehicle-treated group. Bar graphs are means ± SD.

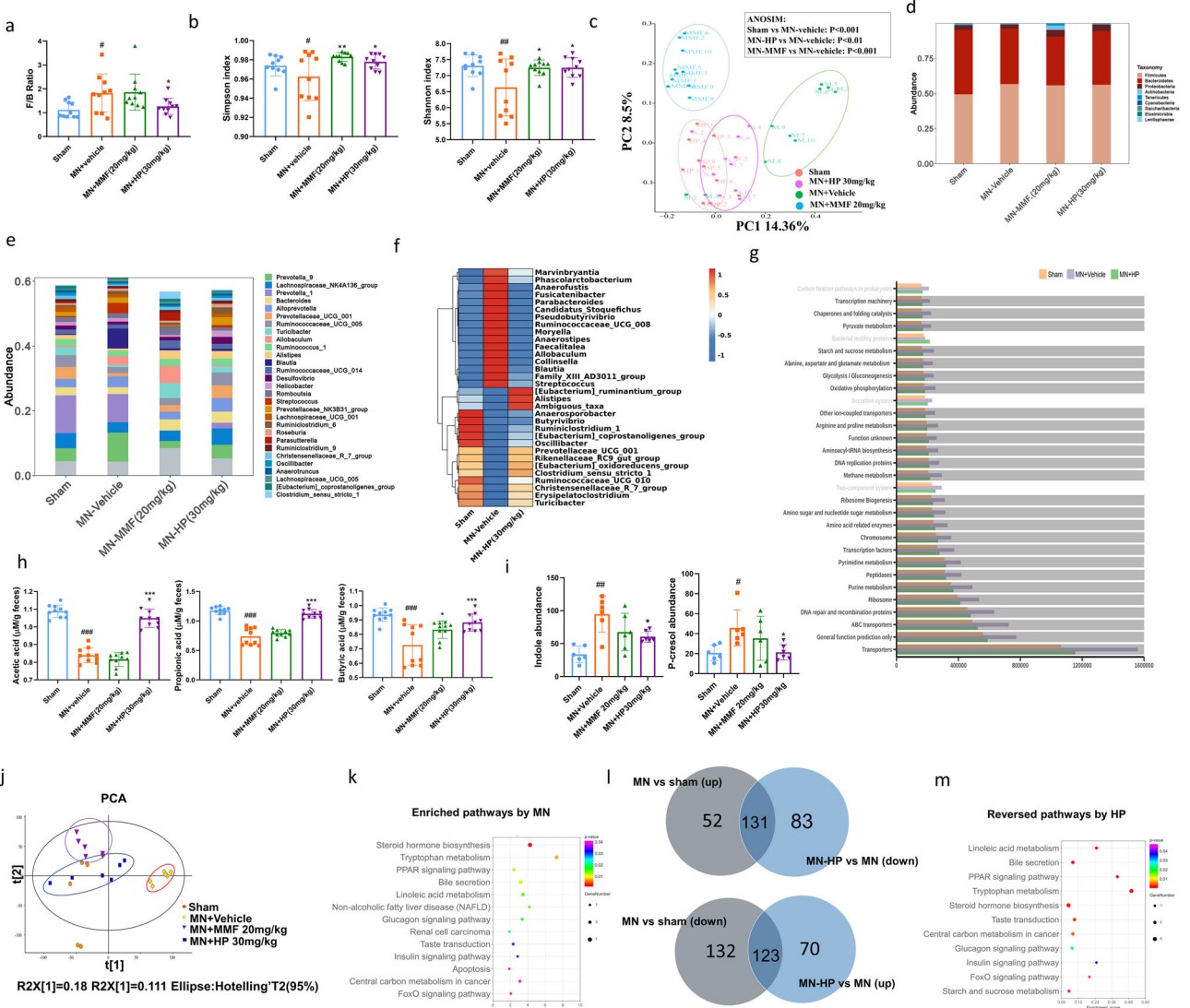

**Fig. 2 Gut dysbiosis is normalized by HP. a** F/B ratio; (n = 10). **b** shannon and simpson index; (n = 10). **c** PcoA analysis among sham, MN+vehicle, MN + MMF and MN + HP (30 mg/kg) groups using the Bray–Curtis distance matrix; (n = 10). **d** taxonomic distribution on phylum level among different groups; (n = 10). **e** taxonomic distribution on genus level among different groups; (n = 10). **f** Top 30 differential genera caused by MN reversed by HP treatment; **g** The extent of KEGG pathway and module enrichment in the gut microbiota of the different experimental groups. The pathways disturbed by MN but restored by HP were shown in bold dark; (n = 10). **h** Concentration of acetic acid, propionic acid, and butyric acid in feces; (n = 6). **i** Concentration of indole and p-cresol in feces; **j** PcoA analysis of metabolite composition among sham, MN+vehicle, MN + MMF and MN + HP (30 mg/kg) groups; **k** KEGG pathways enriched by differential metabolites caused by MN; **l** Differential metabolites caused by gut dysbiosis of MN reversed by HP treatment; **m** KEGG pathways enriched by HP reversed metabolites in MN rats. Significant differences are indicated: #P < 0.05, ##P < 0.01, ###P < 0.001 versus sham group. *P < 0.05, **P < 0.01, versus vehicle-treated group by one-way ANOVA test, (n = 6). Bar graphs are means ± SD.

infiltration of immune cells into kidney interstitium (Supplementary Figs. S3 and S5), HP had capability to alleviate the renal inflammation in MN rats.

**HP normalizes the MN-induced dysbiosis of intestinal microbiome and fecal metabolomics.** The feces from animals in sham, MN-Vehicle, MN-MMF and MN-HP 30 mg/kg group were collected for 16s rDNA sequencing (N = 10), and total 3585 OTUs were identified (BioProject ID:PRJNA1025330). The dysbiosis was commonly characterized by increased ratio of Firmicutes-to-Bacteroidetes. From the data, we could observe thatFirmicutes-to-Bacteroidetes (F/B) ratio was remarkably elevated in MN-vehicle group compared with sham group, but reduced by HP treatment (Fig. 2a), while MMF treatment did not decrease F/B ratio (Fig. 2a). Chao1 and observed species indices were used for

evaluation of richness of gut microbiota, and Shannon and Simpson indices were used to determine the microbial diversity, and we found that both richness and diversity were remarkably reduced in the MN-vehicle group, but up-regulated by HP and MMF intervention (Supplementary Fig. S6 and Fig. 2b), and HP had slightly better effect than MMF on increasing the richness (Supplementary Fig. S6). Principal coordinates analyses (PCoA) revealed that the microbe population between sham and MN-vehicle groups were distinctly separated into two groups (Fig. 2c), and clear separations were also observed for MN-vehicle vs. MN-MMF and MN-vehicle vs. MN-HP 30 mg/kg groups (Fig. 2c). The microbes in MN-sham and MN-HP groups were more overlapped clustered relative to other groups, which was an indication that HP treatment shift microbial composition to the similarity with sham group (Fig. 2c). Relative abundances

of 9 taxomic phyla were shown in Fig. 2d, and all the phyla changes caused by MN were reversed by HP treatment. On the genus level, representative top 30 genera with higher abundance in four groups were shown in Fig. 2e, and most taxonomic levels altered by MN had been reversed by HP. By Kruskal Wallis analysis, compared to sham group, total 35 genera had been significantly changed by MN, and 34 of them were reversed by HP, which was observed in Fig. 2f.

KEGG was used to reveal the pathway enrichment. The result demonstrated that some pathways involved in metabolism were mainly enriched in the MN-vehicle mice, including pyruvate, alanine, glutamine, aspartate, arginine, purine, pyrimidine, and glycolysis and gluconeogenesis. Notably, 27 of 30 KEGG pathways affected by MN were reversed by HP treatment (Fig. 2g, unchanged pathways in light gray). Interestingly, genera related to SCFA generation were found enriched in HP group, such as Clostridum-sensu-stricto and Prevotellaceae-UCG-001 for acetic acid[11], Alistipes and Oscillibacter for propionate acid[12,13], and Rumino-coccaceae-UCG-010, Eubacterium and Butyrivibrio for butyric acid[14]. Consistently, HP treatment increased these SCFAs levels in the feces of MN rats (Fig. 2h), although larger variance of concentration of different SCFAs existed in some of the groups, e.g., MN+vehicle, especially for butyric acid. As shown in Fig. 2f, HP treatment reduced the abundance of Allobaculum and Blautia, which have been reported enriched in CKD patients and both of them played a key role in uremic toxins production[15,16]. Therefore, we further screened the fecal concentration of two typical uremic toxin precursors, indole and p-cresol in metabolomics data. Consistent with the alteration in intestinal microorganism, the accumulation of indole and p-cresol were significantly reduced by HP treatment, however, MMF treatment did not show this similar effect (Fig. 2i).

Global metabolomics alteration is reflective of gut microbiota dysbiosis, and the enriched function signaling pathways derived from differential metabolites could predict the alteration of microbiota biofunctions. Concerning the present findings, by untargeted metabolomics, as demonstrated in Fig. 2j, metabolomics cluster of four groups were separated clearly, except sham group and HP-treated group were partly overlapped, which demonstrated that HP treatment normalized the MN-induced metabolomics dysbiosis, consistent with modulation of gut microbiota. Compared with sham group, in the MN-vehicle group, there were total significant 183 up-regulated and 155 down-regulated metabolites (Supplementary Fig. S7, VIP > 1, $P < 0.05$). KEGG enrichment analysis using these differential metabolites revealed that there were total 14 significantly altered pathways ($P < 0.05$), including PPAR signaling pathway, steroid hormone biosynthesis, tryptophan metabolism and et al. (Fig. 2k). By Venn diagram, as shown in Fig. 2l, 131 of 183 up-regulated and 123 of 155 down-regulated fecal metabolites were significantly reversed by HP administration, and these reversed metabolites were enriched in 12 of 14 altered KEGG pathways altered by MN (Fig. 2m). All these results demonstrated that HP treatment could reversed the abnormal metabolites caused by microbiota dysbiosis.

**HP restores gut immunity and reinstates gut integrity.** Gut dysbiosis induced the dysregulation of immune system and destroyed the homeostasis of gut immune microenvironment[17]. As shown in Fig. 3a, MN-induced dysbiosis triggered increased infiltration of lymphocytes and monocytes into colon tissues, and HP oral administration could suppress these phenomena. In detail, HP administration reduced the proportion of proin-flammatory Th1 (CD4 + IFNγ + ) and Th17 (CD4 + IL17A + ) cells, as well as M1 (CD68 + CD86 + ) macrophages in the colon

tissues. Meanwhile, HP intervention also restrained the production of inflammatory cytokines, including TNFα, IL1β and IL6 in colon tissues (Fig. 3b).

Next, we examined whether HP protected intestinal integrity and morphological structure. In vehicle-treated MN group, we observed villi necrosis, goblet cells reduction, edema and ulceration in intestinal tissues by HE staining, and the macroscopic injury score was much higher in MN rats compared with sham control, while HP significantly reversed such these pathological injuries (Fig. 3c). Notably, indistinct tight junction, reduced intestinal villi, and damaged desmosome structure in intestinal tissues of MN rats was found in TEM observation, and HP treatment also ameliorate such detrimental changes (Fig. 3c).

C-BSA challenge remarkably decreased expression of the intestinal epithelial tight junction proteins, including claudin-1, mucin 2, ZO-1 and occludin, and HP treatment could reverse these effects, which was confirmed by western blot (Fig. 3d). All these findings supported that HP administration might reinstate intestinal barrier integrity in experimental MN rats.

CKD impairs gut permeability and leads to release of bacterial lipopolysaccharides (LPS) into the circulation[18]. Endotoxemia is one of important causes of production of proinflammatory cytokines in peripheral circulation and kidney, which results in chronic inflammation in c-BSA induced MN. We inspected the effects of HP on LPS levels in serum and kidneys, and results confirmed that HP could reduce circular endotoxemia as well as LPS in kidney tissues (Fig. 3e), however, the MMF did not reduce the LPS concentration both in serum and kidneys effectively.

**Therapeutic effect of HP on MN was transmissible by FMT.** To prove that gut microbiota plays a decisive role in the renal protective effects of HP, we applied FMT by gavage recipient c-BSA challenged MN mice with fecal samples extracted from experiment 1, sham (R-sham), MN-vehicle (R-MN), MN-MMF treated (R-MMF) and MN-HP-treated (R-HP) donor rats. The feces from receipt rats were collected for 16s rDNA sequencing (BioProject ID: PRJNA1025250). The detailed experimental procedure was shown in Fig. 4a. As shown in Fig. 4b, lower gut bacteria abundance and diversity were observed in the recipient rats in the R-MN group (Chao1, observed species, Shannon and Simpson index) compared to those in the R-sham group; however, these changes were completely reversed in R-HP rats, but partially in R-MMF rats (R-MMF group had significantly higher Chao1 and observed species but not Simpson and Shannon index compared with R-MN group). PCoA analysis showed that the overall gut microbiome structure of R-MN rats was quite different from that of the R-sham rats. However, the difference is reversed in R-HP rats, but not in R-MMF (Fig. 4c). We examined the taxonomic composition of the recipient gut microbiome and observed that gavage with feces of MN rats had higher F/B ratio at the phylum level, while feces from HP donors restored these changes (Fig. 4d), but feces from MMF donor could not reverse this phenomenon. The composition changes of gut microbiota in recipient rats were consistent with those in donor rats (Fig. 4e). Subsequently, the intestinal morphological structure was observed by HE staining and TEM, and the intestinal integrity in R-HP rats was better than in R-MN rats, also better than in R-MMF group. As determined in Fig. 4f, g, MN rats in the R-MN group had fewer goblet cells, micro villi and epithelial tight junction proteins than the R-sham group. FMT with feces of HP-treated rats prevents this consumption and maintains normal conditions (Fig. 4f, g). Consistent with impaired gut integrity, increased plasma and kidney LPS levels were higher in R-MN rats than R-sham rats, and were restrained by intervention with feces from HP-treated rats (Fig. 4h, i).

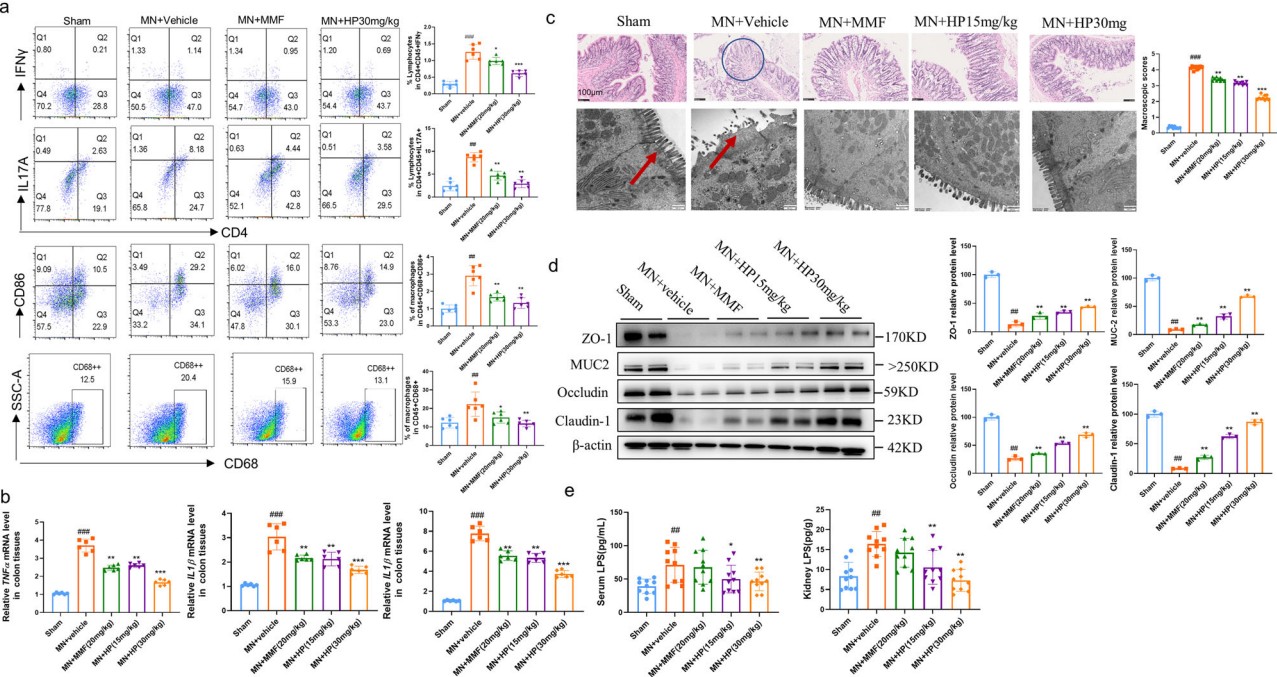

**Fig. 3 HP reduces endotoxia and protect the intestinal tight junctions in MN rats. a** Representative flow cytometric profiles of M1 macrophages, Th1 and Th17 cells in gut tissue. Th1 cells were gated as CD4+/INFγ+ cells in CD45+lymphocytes; Th17 cells were gated as CD4 + /IL17+cells in CD45+lymphocytes; M1 cells were gated as CD68 + CD86+ cells; (n = 6). **b** Expression of *IL6, IL1b,* and *TNFα* mRNA in gut tissues were evaluated by RT-PCR. The results were normalized to GAPDH. (n = 10). **c** representative pathological photographs of light microscopy (H&E) and electronic microscopy for intestinal tissues; red arrows refer to intestinal villi; (n = 3, black bars = 100 µm, white bars = 500 nm). **d** protein levels of tight junction protein in intestinal tissues. (n = 10). **e** LPS content in serum and kidney. #$P < 0.05$, ##$P < 0.01$, ###$P < 0.001$ versus sham group. *$P < 0.05$, **$P < 0.01$, versus vehicle-treated group by one-way ANOVA test. Bar graphs are means ± SD.

FMT with feces from HP rats also ameliorated c-BSA challenged kidney injuries, which was demonstrated by reduced albuminuria and serum cholesterol (Fig. 4j), as well as decreased glomerular IgG deposition (Fig. 4k). These results indicated that the benefits of HP could be at least partially attributed to its effect of gut microbiota. However, compared with oral administration of HP, fecal transplant was less effective in reducing the BUN and Scr (Fig. 4j), which also suggested that gut microbiota modulation by HP only contribute partially to its renal protective effect, and the remaining beneficial effect was derived other pharmacological effect, which was early reported by our previous studies[4].

**Depleted gut microbiota remarkably abolishes the renal protective of HP against c-BSA induced MN in rats**. To further confirm whether modulation of gut microbiota by HP contributed to its renal protective effect on MN rats, we prepared an antibiotic cocktail in drinking water to deplete the gut commensal bacteria in c-BSA induced animals before HP administration. More than 90% of bacteria from the intestine were cleared by this antibiotic mixture, and we confirmed that by counting the bacterium colonies in streak culture using feces from animals (Supplementary Fig. S8). As shown in Fig. 5b, antibiotic treatment remarkably abolished the therapeutic efficacy of HP, which was indicated by that HP less effectively decreased albuminuria, serum NGAL and serum cholesterol in pseudo germ-free animals compared with wild-type MN animals, even did not decrease the BUN and Scr in pseudo germ-free MN rats. IgG deposition in glomeruli is one of important pathological characteristics in MN, and as shown in Fig. 5c, HP treatment can reduce the serious glomerular IgG deposition in wild type MN rats notably, but this inhibitory effect was remarkably diminished due to depletion of gut microbiota, consistent with biochemical data of renal function. All these results indicated that

gut microbiota is required to booster the therapeutic effect of HP against experimental MN in vivo.

**Major coumarins in HP are bio-transformed into 7-hydroxycoumarin in ex vivo gut microbiota and reduces the uremic toxin production**. The existence of gut microbiota is necessary for HP's renal protective effect, which indicated that HP not only could remarkably restore the gut microbiota dysbiosis, but also the live intestinal bacteria may be possible to chemically transform or modify the compounds in HP, thus to enhance their bioactivities. The previous pharmacokinetics study of oral administration of HP indicated that the coumarin derivates in HP could be metabolized into 7-HC in rats, which had higher drug concentration in both plasma and kidneys with higher bioactivities[4]. However, whether 7-HC is metabolized by hepatic enzymes or gut microbiota, is not clear yet. To preliminarily determine possible metabolic pathways of 7-HC, major compounds in HP, skimmin or apiosylskimmin were incubated with human and rat liver microsomes (LM) as the substrate. Results showed that the positive control midazolam a substrate of cytochrome P450 enzymes (CYPs) was reduced by 98% in the presence of nicotinamide adenine dinucleotide phosphate (NADPH) in human and rat LMs, indicating that the incubation system was responsible. However, apiosylskimmin (Fig. 6a) and skimmin (Fig. 6b) were stable in human and rat LMs, which suggested that CYPs were unlikely to participate in the metabolism of skimmin and apiosylskimmin into 7-HC.

Subsequently, we investigated whether gut microbiota was involved in the biotransformation of apiosylskimmin and skimmin. Our data showed that skimmin and apiosylskimmin was unstable in rat gut microbiota. When skimmin was incubated with gut microbiota, it was time-dependently metabolized to 7-HC, and the

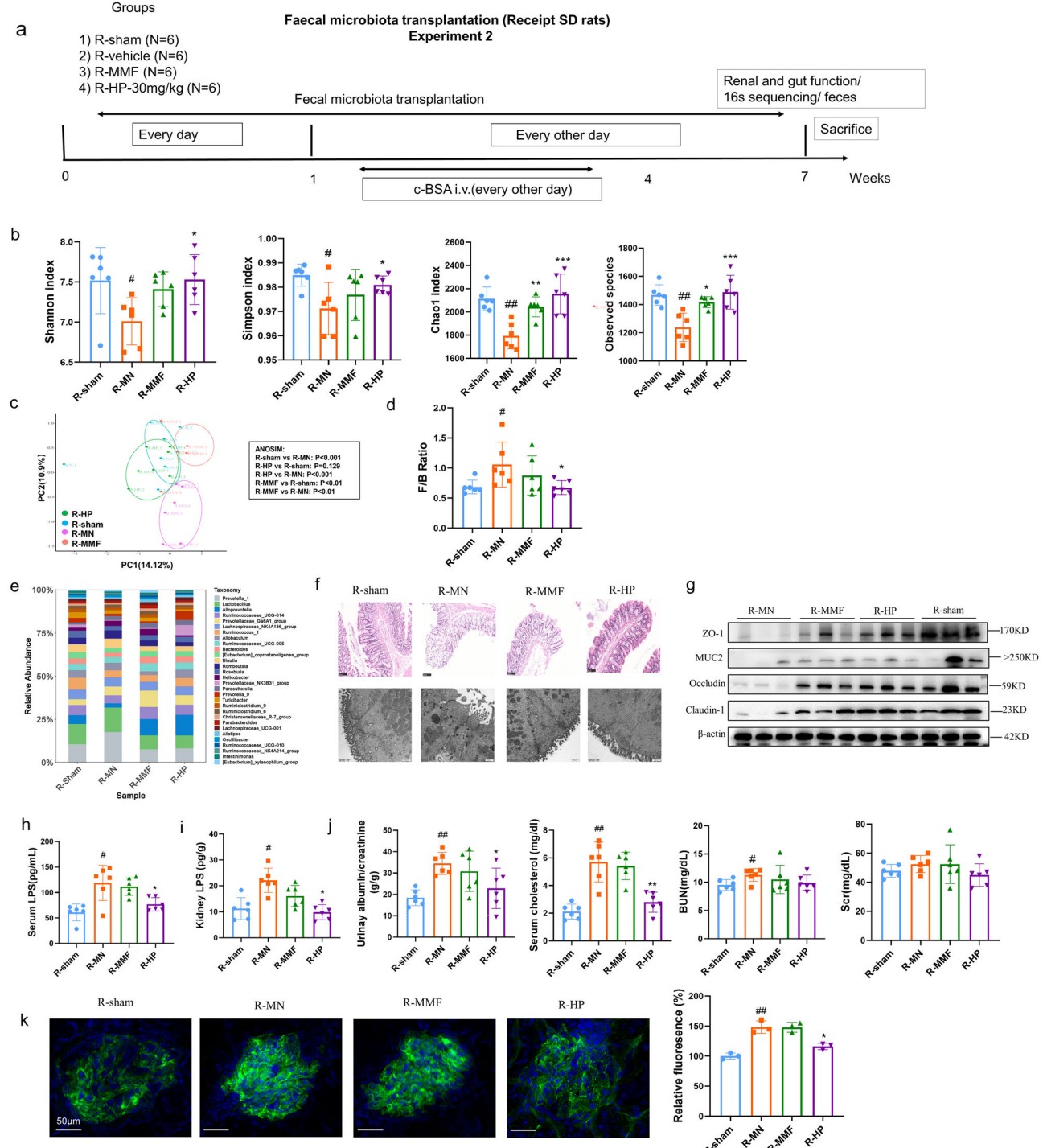

**Fig. 4 Fecal transplant using HP-treated feces lowering albuminuria and protects the intestinal integrity in MN rats. a** Scheme of fecal transplant; (n = 6). **b** Shannon, Simpson, Chao 1and observed species index among four FMT groups; (n = 6). **c** PcoA analysis among four FMT groups; (n = 6). **d** F/B ratio among four FMT groups; (n = 6). **e** relative abundance of several probiotics and conditional pathogens on genus level among FMT groups. **f** representative figures of intestinal tissues by HE staining and TEM;(Black bars = 100 μm, white bars = 500 nm). **g** tight junction protein expression determined by western blot; (n = 6). **h** LPS content in serum; (n = 6). **i** LPS content in kidney. (n = 6). **j** FMT using HP-treated feces reduces the albuminuria and serum cholesterol, but not BUN and Scr; (n = 6). **k** FMT using HP-treated feces reduces the IgG deposition in glomeruli in MN rats. #P < 0.05, ##P < 0.01, ###P < 0.001 versus R-sham group. *P < 0.05, **P < 0.01, versus R-MN group by one-way ANOV A test. (White bars = 50 μm).Bar graphs are means ± SD.

amount of generated 7-HC was close to the reduced skimmin. Two hours later, most skimmin were metabolized into 7-HC (Fig. 6c). After incubating with gut microbiota for 0.5, 1, 2 h, apiosylskimmin decreased by 17.1%, 20.0% and 37.4%, respectively, and converted to skimmin and 7-HC over time (Fig. 6d). From the Fig. 6d, we can hypothesize that apiosylskimmin was firstly metabolized into skimmin, then skimmin continued to be metabolized into 7-HC by gut microbiota. All these results indicated that main resource of 7-HC in vivo was from the gut microbiota metabolism instead of liver metabolism. Further, $H_2O_2$-induced ROS assay using

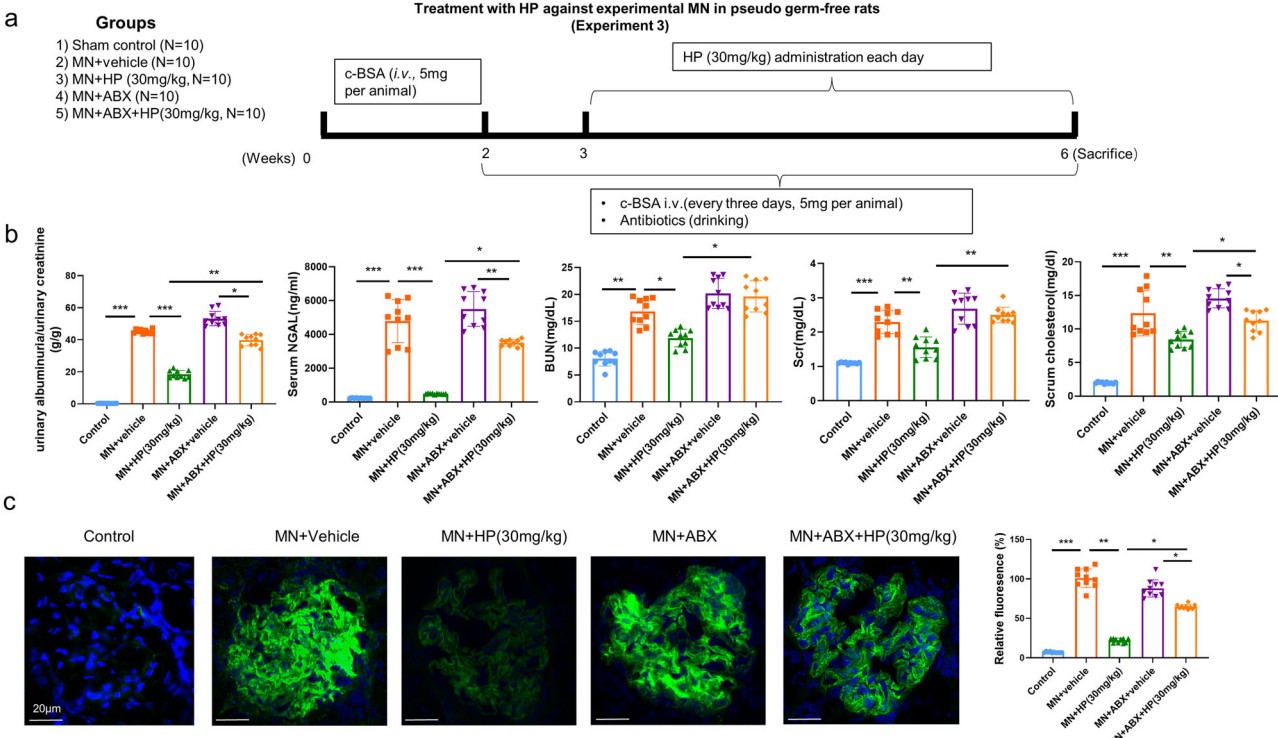

**Fig. 5 Depleted gut microbiota remarkably abolishes the renal protective of HP against c-BSA induced MN in rats. a** Illustration of experimental design to treat MN rats whose gut microbiota have been depleted by ABX drinking; ($n = 10$). **b** Renal function, albuminuria, serum cholesterol in different groups; ($n = 10$). **c** Reduction of IgG deposition by HP was partially diminished by gut microbiota depletion via immunofluorescence. *$P < 0.05$, **$P < 0.01$, tested by one-way ANOVA test. (White bars = 20 μm). Bar graphs are means ± SD.

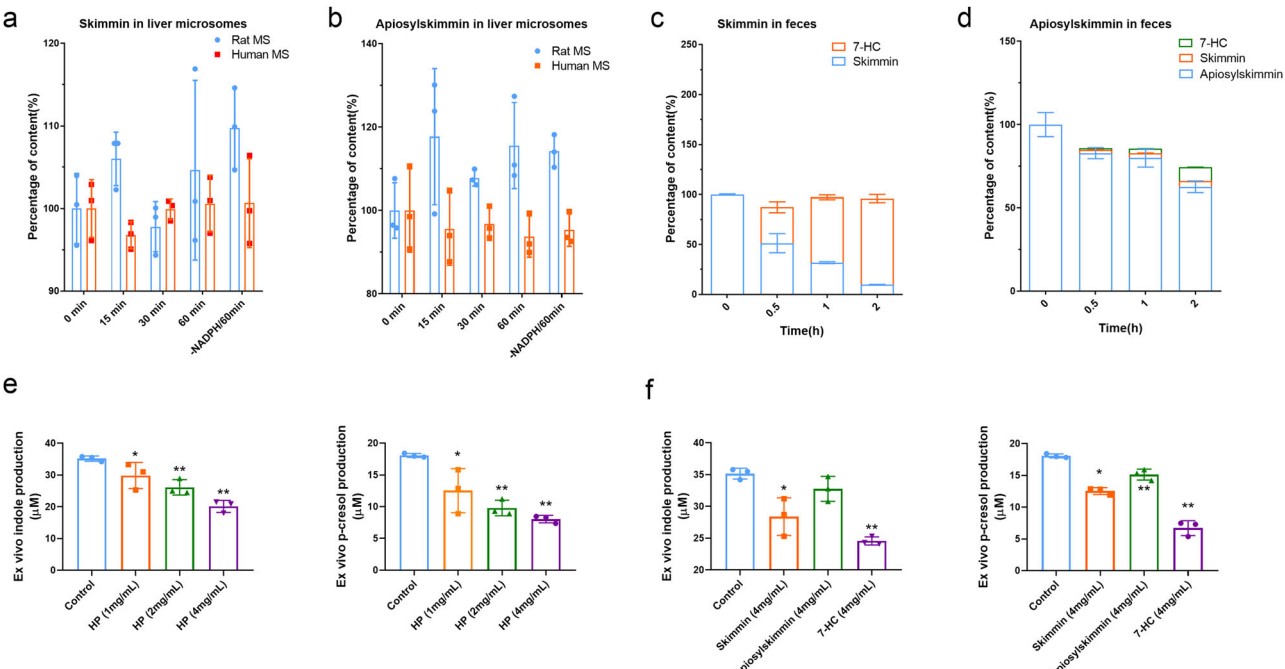

**Fig. 6 Major coumarins, skimmin and apiosylskimmin, were metabolized into 7-HC by gut microbiota, and reduces the indole and p-cresol production ex vivo. a** Metabolism characteristics of apiosylskimmin in feces at different time points; ($n = 3$). **b** Metabolism characteristics of skimmin in feces at different time points; ($n = 3$). **c** Metabolism characteristics of apiosylskimmin in liver microsome at different time points; ($n = 3$). **d** Metabolism characteristics of skimmin in liver microsome at different time points; ($n = 3$). **e** HP reduces the production of indole and p-cresol in ex vivo feces; ($n = 3$). **f** 7-hydroxylcoumrin has the highest inhibitory effect on production of indole and p-cresol in ex vivo feces than skimmin and apiosylskimmin. *$P < 0.05$, **$P < 0.01$, tested by one-way ANOVA test. Bar graphs are means ± SD.

mesangial cells showed that 7-HC has more anti-oxidation bioactivity than skimmin and apiosylskimmin (Supplementary Fig. S9). All these results indicated that there was bi-directional interaction between HP and gut microbiota, and enhanced kidney beneficial effect of HP in vivo.

Finally, we tried to confirm that HP could reduce the uremic toxin precursor production of gut microbiota in ex vivo feces, which were found by fecal metabolomics study. The ex vivo indole and p-cresol production were examined by cecal content culture in anaerobic medium, and results proved that HP reduced their production significantly dose dependently (Fig. 6e). The positive control 4-nitrophenyl-β-D-glucopyranoside (200 μM) was reduced by 99% after incubation with intestinal bacteria, indicating that the incubation system was reliable.

Moreover, we investigated which exact chemical compounds in HP played such role. As shown in Fig. 6f, all skimmin, apiosylskimmin and 7-HC could reduce the indole and p-cresol production within six hours, but 7-HC had highest inhibitory effect, and apiosylskimmin was lowest. Combined with fecal metabolism data from Fig. 6c, d, we may speculate that the effect of skimmin reducing the indole and p-cresol production was derived from that skimmin was biotransformed into 7-HC by microbiota. The possible reason why apiosylskimmin had lowest effect was due to its slower speed of 7-HC biotransformation compared with skimmin. And we also could make a hypothesis that HP reducing uremic toxin precursors might be from 7-HC production due to gut microbiota existence.

The up-regulated richness and diversity of gut microbiota in MN rats by HP suggested that HP did not inhibit the gut microbiota growth, which indicated that reduction of uremic toxin was not from inhibiting bacteria proliferation. HP might inhibit some key enzymes activities in gut microbiota, which may be involved in the fermentation and indole and p-cresol production. KEGG pathway analysis (Fig. 2g, m), for example, tryptophan metabolism inhibition (indole production way) by HP also provided the clues but deserves further study.

## Discussion

Previous research from our laboratory has confirmed the therapeutic effect of HP on c-BSA-induced MN in rats, and its possible mechanisms include anti-inflammation by inhibiting activation of the complement and ameliorating fibrosis by blocking the TGFβ1-smad3 signaling pathway[4]. However, because HP is a mixture of coumarin derivatives, its exact pharmacological mechanism is still not very clear. Recently, accumulating evidence has suggested that the development and progression of CKD involves gut microbiota dysbiosis, which means that a therapeutic strategy against CKD based on regulating the gut microbiota is feasible[19]. On the other hand, additional data suggest that the medicinal effects of many herbal medicines occur through modulating the gut microbiota; therefore, it is reasonable to speculate that the reno-protective effect of HP might be partially mediated by the restoration of gut symbiosis[20,21]. In addition, for natural compounds with lower oral bioavailability, biotransformation and chemical modification by the gut microbiota may lead to the production of more bioactive compounds with higher bioavailability. In the current study, with the help of 16S rDNA sequencing, metabolomics and fecal transplantation, we confirmed this hypothesis.

Serious gut dysbiosis occurs in MN rats, and this dysbiosis is characterized by a higher F/B ratio and reduced diversity and richness. HP reverses these shifts in the gut microbiota, and its renal protective effects are transferrable partially through fecal transplantation, which supports the idea that alterations to the gut microbiota are involved in CKD. These findings are consistent with several previous results[22], which demonstrated that

gut leakiness and early renal injury could be triggered independently by transferring the gut microbiota of CKD individuals[6] or animals to germ-depleted mice. Our results suggest that HP administration or fecal transfer can change the gut microbiota and may be used as potential prebiotics to modulate the gut microbiota composition; this modulation might be associated with maintaining gut barrier integrity and reducing uremic toxin precursor production. Although in the current study, FMT was performed in rats containing intrinsic gut microbiota, the taxonomic analysis by 16S rDNA sequencing, PCoA analysis and α-diversity showed that the composition of gut microbiota in recipient animals was changed to a profile similar to that of donor rats, which demonstrates the successful establishment of FMT.

The results of the present study implied that HP administration could "normalize" gut microbiota dysbiosis, which is different from the effects of MMF treatment. MMF treatment also distinctly alters the gut microbiota composition, but the newly formed microbiota is remarkably different from that of sham rats. The difference in the microbiota was determined based on the clear separation of the clusters by PCoA analysis. The F/B ratio is the most widely accepted gut dysbiosis biomarker with reference to chronic inflammation and other pathological statuses, and only HP administration could reduce this ratio. The results of our study suggested that the diversity of the gut microbiota of rats was enhanced by oral administration of HP, and HP administration could regulate the relative abundance of several key bacterial species reported to be associated with CKD development. At the genus level, the abundance of Desulfovibrio, Blautia and Streptococcus increased remarkably in CKD patients in different cohort studies[23], as well as in the current MN rat model. Desulfovibrio can aggravate dysfunction of the gut mucosal barrier by extending the inflammatory profile and damaging colonocytes with $H_2S$[24]. Streptococcus and Blautia are related to uremic toxins such as indole derivates and p-cresol and have been contrarily associated with kidney function (eGFR). HP administration triggered a significant reduction in the abundance of these genera. Otherwise, the abundance of known commensal bacteria, including Prevotellaceae_UCG_001, Roseburia and Bifidobacteria, which were reduced among kidney disease populations, are related to improved kidney function (eGFR) and reductions in cystatin C levels, BUN and Scr[25]. In the current study, the lower abundance of these commensal bacteria was significantly reversed by HP administration. All these results provide evidence and background for exploring the mechanism of HP from the aspect of the gut microbiota.

Over the past few decades, there has been an increasing amount of research concerning the role of chronic systemic inflammation in the progression of CKD. Decreased intestinal barrier function and increased intestinal permeability are important intestinal changes in patients with CKD[26]. As a result of altered occludin and claudin expression, the translocation of gut microbiota and/or endotoxins into the circulation through disruption or leakage at colonic epithelial tight junctions might also trigger systemic inflammation[27]. In the current study, remarkable impairment of the intestinal barrier was observed by histopathology, and further elevated serum LPS, possibly caused by the translocation of pathogenic bacteria, was also confirmed in the MN-vehicle group. Following HP treatment, the intestinal epithelial integrity was repaired, and local colon tissue inflammation, inflammatory cell infiltration, and endotoxemia were significantly ameliorated. Although less effective than HP treatment, fecal transplantation using HP-treated rats also reduced MN-induced intestinal permeability impairment, and consequently reduced serum endotoxemia. These results confirm that gut microbiota modulation by HP contributes to the protection of

gut barriers and restricts the translocation of invading bacterial pathogens based on intestinal permeability.

Metabolomic results demonstrated that HP treatment reduced the production of fecal uremic toxins, including conjugated indole derivates and p-cresol glucuronide, especially indole metabolites (Supplementary Fig. S10). Increased abundance of *Allobaculum, Desulfovibrio*, and *Enterorhabdus*, which are candidate bacteria that can produce uremic toxins in the rat intestinal tract, was associated with CKD severity[15]. The decrease in the relative abundance of these bacteria by HP partially explains the lower levels of toxins. In the gut, the tryptophan metabolism pathway, which produces indole derivatives, is under the direct control of the microbiota[28], and by KEGG pathway analysis, HP treatment significantly downregulated tryptophan metabolism, which was one of the possible underlying mechanisms of HP treatment. SCFAs, such as acetate, propionate and butyrate, can nourish cells, have anti-inflammatory effects against chronic inflammatory diseases, and promote colonocyte health[29]. Based on metabolomics analysis, the levels of these SCFAs were significantly increased by HP treatment, which is consistent with the increase in SCFA-producing bacteria, such as the genera *Roseburia, Prevotellaceae-UCG-001, Lactobacillus* and *Bifidobacterium*, by HP treatment; these bacteria are intestinal probiotics that play an important role in improving the intestinal microenvironment and maintaining intestinal health. These results indicate that HP could not only restore the gut microbiota of MN rats to a composition similar to that of sham rats but also increase the abundance of beneficial bacteria and reduce that of conditional pathogens; thus, HP seemingly acts as a potential prebiotic for CKD patients.

Meanwhile, through hydroxylation and reduction, 7-HC, a bioactive metabolite of HP in vivo, has been investigated by pharmacokinetic studies[2,4]. Although skimmin and apiosylskimmin are major chemical compounds in HP, 7-HC had higher blood drug concentrations in plasma and abundant accumulation in kidney tissues. 7-HC may contribute more to the therapeutic effect of HP against CKD than skimmin and apiosylskimmin. 7-HC was widely reported to show beneficial effects in both experimental acute kidney injury[30] and CKD[31,32], mainly through antioxidation and NFκB signaling inactivation[33]. By incubating liver microsomes and gut bacteria with skimmin and apiosylskimmin, we demonstrated that 7-HC is derived from the gut microbiota instead of liver enzymes. From this perspective, the gut microbiota is necessary to improve the pharmacological effects of HP in vivo, as the gut microbiota improves the bioavailability of bioactive metabolites, in contrast with the bioavailability of the parent molecules skimmin and apiosylskimmin. Our current study is a typical case of a bidirectional interaction between natural compounds and the gut microbiota, which induces synergistic effects in vivo. Furthermore, we confirmed that 7-HC contributes to decrease in indole and p-cresol by ex vivo fecal metabolism research. Partial abolishment of the kidney protection effect of HP by depletion of gut microbiota also provides evidence that 7-HC transformation by the gut microbiota contributes to its renal protection effect. The renal protection mechanism of HP based on gut microbiota modulation is summarized in Fig. 7.

There are many limitations to the current study that need to be mentioned. First, the rodent model of c-BSA-induced MN utilized in this study has a microbiota composition distinct from that of patients in many aspects, so these findings need to be carefully interpreted while providing implications for future human studies. Second, the detailed molecular mechanisms by which HP and 7-HC reduce indole and p-cresol production deserves further study, as well as which bacterial enzymes, such as tryptophanase, that are involved in this process.

## Conclusions

In summary, our findings indicated that HP could normalize the dysbiosis of the gut microbiota, which contributed to its beneficial effect in experimental MN; furthermore, 7-HC was bio transformed by the gut microbiota from skimmin and apiosylskimmin, which increased HP bioavailability and enhanced its pharmacological effect in vivo.

## Materials and methods

**Establishment of c-BSA induced experimental MN and HP administration**. HP was prepared by the State Key Laboratory of Bioactive Substances and Functions of Natural Medicines, Institute of Materia Medica, Chinese Academy of Medical Sciences[34]. A high-performance liquid chromatography-based chemoprofile of HP has been shown in Supplementary Fig. S11. The cationized BSA was prepared using methods described by Border[35,36]. Capillary Isoelectric Focusing facilitated by 111 IEF Cell (Bio-Rad, CA, USA) was used to examine the isoelectric point (PI) of BSA. The PI confirmation of c-BSA was shown in Supplementary Fig. S12.

Female Sprague Dawley (SD) rats, weighing 180−220 g, were obtained from the Institute of Laboratory Animal Science, Chinese Academy of Medical Sciences (Beijing, China). Experimental MN model was induced by tail vein injection of 5 mg c-BSA per animal for consecutive 14 days. Another ten animals received saline via tail vein injection as sham control. Then urine was collected to examine the albumin concentration using rat urine albumin kit (abcam, Cambridge, MA, USA). Urine albumin/creatinine ratio 10 times higher than sham control was regarded as the standard for model success. Animals with albuminuria were then randomly divided into four groups for future drug administration vehicle, MMF (20 mg/kg), and HP (15 and 30 mg/lg). HP and MMF was dissolved in 0.5% carboxymethyl cellulose sodium (CMC-Na) and orally given once daily for consecutive six weeks. MN-vehicle group and sham control group received same volume of solution buffer orally. The detailed grouping and treatment procedure was shown in Supplementary Fig. S13. The rationale of dose selection and treatment duration was adhered to the previous publications, and based on balance between drug efficiency and toxicity[1,4]. All animal experiments were approved by the Ethics Committee of Laboratory Animals of the Peking Union Medical College in Beijing, China (approval number 002981, August 2019). After all the animals were sacrificed, the kidney and colon tissue were collected for pathological observation. This experiment was termed as experiment 1.

**Evaluation of albuminuria, lipidemia, endotoxia and renal function**. Before euthanasia, the blood from each animal were collected through the eyes for the biochemical test by automatic biochemical analyzer (HITACHI 7600, Tokyo, Japan), including BUN, serum creatinine (Scr), serum triglyceride, serum total cholesterol. Serum NGAL was examined by enzyme-linked immunosorbent assay (Enzyme-linked immunosorbent assay (ELISA)) kit (abcam). Serum LPS was measured by ELISA using luminometer (BioTek Instrument, Inc., Vermont, CA, USA) at 450 nm, according to the manufacturer's instructions (LSBIO, Seattle, WA, USA).

**Light microscopy and electron microscopy**. Paraffin-embedded kidney and colon sections which were stained with hematoxylin and eosin (HE) or Masson's Trichrome (Accustain, Sigma, St Louis, MO) were observed under light microscopy. Image analysis software (NDP Viewer 2; Hamamatsu Photonics, Tokyo, Japan) was used to further analyze the images of panoramic scanning.

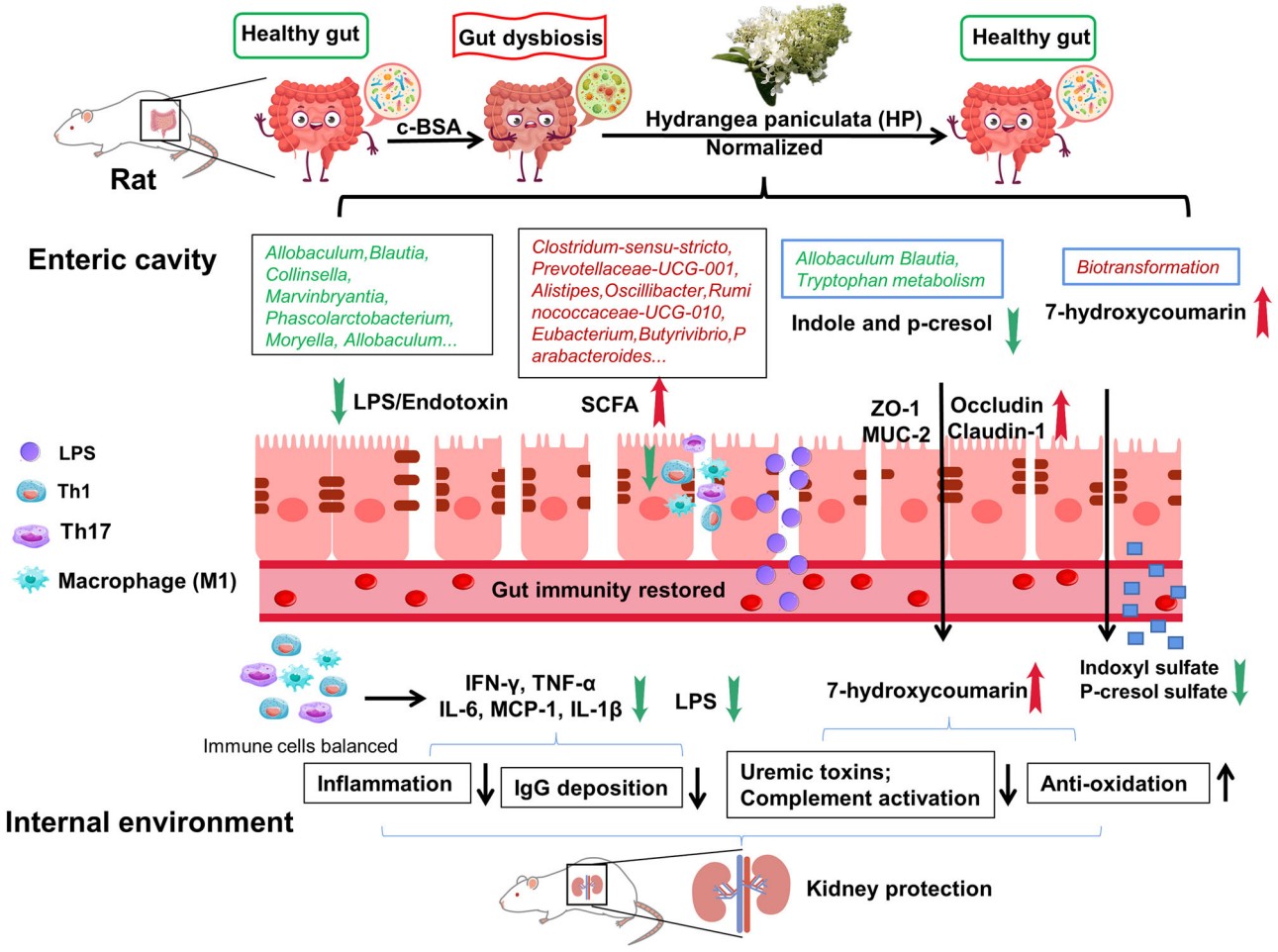

**Fig. 7 summary of renal protection mechanisms of HP through bidirectional interaction with gut microbiota in c-BSA induced experimental MN in rats.** HP effectively modulates the composition, function, and production of endogenous metabolites of gut microbiota. The gut health reinstated by HP benefits systemic immune cell dynamics and kidney functions, leading to improved chronic inflammation and glomerular IgG deposition. Consequently, renal local inflammation, macrophage infiltration, uremic toxin production and albuminuria attenuated, causing less loss of kidney function.

The detailed scoring methods for glomerular and tubular injury was described in the supplemental methods.

Besides of light microscopy, TEM was conducted for observing kidney and colon ultrastructure. The ultrathin sections were prepared. Briefly, small pieces of the left kidneys and colon tissues (1 × 1 mm) were fixed in 2.5% glutaraldehyde solution buffered with sodium cacodylate buffer at pH = 7.4 for 2 h, post-fixed for 1 h in 1% osmium tetroxide solution at the same temperature and pH, dehydrated in ethanol. After embedded, sections were stained with uranyl acetate and lead citrate. JEM-1400 TEM (JEOL USA, Inc, Peabody, MA, USA) were used for viewing and photographing. Evaluations were performed by two independent observers in a blinded fashion. Grading for the podocyte deterioration change was assessed according to the following scale: 0 – no evidence of changes; I – <25% changes; II – 25−50%; III – 50−75%; IV->50%. Thickness of the GBM was also measured. In addition, ultrastructure of tight junctions and intestinal villi were also observed in the colon.

**Immunohistochemistry and immunofluorescence**. Immunohistochemical staining was performed following the common procedure. Briefly, kidney tissue sections were subjected to deparaffinization, hydration, and microwave antigen retrieval, followed by incubation at 4 °C for with primary antibody against CD68 (1:100; Abcam) overnight. In negative controls, the primary

antibody was replaced by buffer. After primary antibody incubation, sections were incubated with a peroxidase-conjugated secondary antibody (ZSGB-Bio, Inc, Beijing, China), then was counterstained with hematoxylin. The IgG deposition in glomerular compartment was examined by immunofluorescence. Briefly, frozen kidney sections (5 μm) were treated with a FITC-conjugated goat anti-rat IgG (Sigma Aldrich, St. Louis, MO, USA) for one hour, then the image was collected using a charge-coupled device camera (S610; Hamamatsu Photonics, Tokyo, Japan), as well as immunohistochemistry images. Image analysis software (NDP Viewer 2; Hamamatsu Photonics, Tokyo, Japan) was used to further analyzing the images of panoramic scanning.

**Serum cytokine analysis**. The Rat Inflammation Panel 13-plex (Cat. No. 740251, LegendPlex, Biolegend, USA) was used to measure cytokine levels in serum according to the manufacturer's instructions.

**Reverse transcription and quantitative PCR (qPCR)**. Aims to evaluate inflammation status in kidneys or colon tissues, the mRNA level of several chemokines and cytokines were determined using quantitative reverse transcription polymerase chain reaction (qRT-PCR). Briefly, RNA extraction from kidney cortex tissues was performed using Trizol (life technology, Inc, USA) according to the protocol, and for the colon tissues, we used the

sharp blade to cut the inner layer down from intestinal segments, and to collect more single-layer epithelial cells as much as possible. Reverse transcript kit (Transgen Biotech, Beijing, China) was used to obtain cDNA. The fold change of the genes was observed by running qPCR based on the formula $2^{-\Delta\Delta Ct}$, which used GAPDH as the house-keeping reference gene. The primer sequences could be found in Supplementary table 1.

**Cell isolation, staining and flow cytometric analysis**. As described by Li et al.[11], the intestine of mice was washed with cold PBS and then cut into 1-cm-long segments. Then, the segments were placed in 50 mL centrifuge tube containing 5 mL Hank's balanced salt solution (HBSS) for 20 min at 37 °C. The intestine epithelial cells were removed after vortex for 10 s. The tissue segments were placed into tubes containing 5 mL digestive solution (0.5 mg/mL collagenase D (Roche), 0.1 mg/mL DNase I (Sigma), and 3 mg/mL dispase II (Roche) in DMEM medium) homogenized, and then the supernatant was collected for the flow cytometry staining.

The following antibodies were used for flow cytometry: PE anti-rat CD3 Recombinant Antibody (Biolegend, Cat #200004), APC anti-rat CD4 Antibody (Biolegend, Cat #201509), APC/ Cyanine7 anti-rat CD45 Antibody (Biolegend, Cat #202216), FITC anti-rat CD4 Antibody (Biolegend, Cat #202205), PerCP-Cyanine5.5 IL-17A Antibody (Thermo Fisher, Cat #45-7177-80), FITC CD68 Antibody (Thermo Fisher, Cat #MA5-28262), PE CD86 Antibody (Thermo Fisher, Cat #12-0860-83). Isotype was used for control staining. Six-color fluorescence flow cytometric analyses were performed (FACS Verse, BD, USA), FlowJo_V10 was used to analyze the flow cytometry data.

**Microbiome sample collection and analysis**. Fecal samples from each animal were frozen at −80 °C prior to DNA extraction. Total genomic DNA was extracted using DNA Extraction Kit following the manufacturer's instructions (Invitrogen, CA, USA). Quality and quantity of DNA was verified with NanoDrop and agarose gel. For bacterial diversity analysis, V3-V4 variable regions of 16 S rRNA genes was amplified with universal primers 343 F and 798 R (343 Forward: 5'- TACGGRAGGCAGCAG-3';798 Reverse: 5'-AGGGTATCTAATCCT-3')[37]. For the chimera removal, the tags were compared with the reference database (Silva database, https://www.arb-silva.de/) using UCHIME algorithm (UCHIME Algorithm, http://www.drive5.com/usearch/manual/uchime_algo.html) to detect chimera sequences, and then the chimera sequences were removed. Then the Effective Tags finally obtained. Chao1 and Shannon indices were used to indicate α-diversity, and PCoA was used to indicate β-diversity and was estimated using the Bray–Curtis distance matrix. Phylogenetic investigation of communities by reconstruction of unobserved states was used to infer the predicted functional composition of the gut microbiome of each sample, which was represented by Statistical Analysis of Metagenomic Profiles.

**Untargeted metabolomics and measurement of metabolites**. The profile of metabolites in fecal samples were analyzed by a $2.1 \times 100$ mm ACQUITY 1.8 μm HSS T3 using a Waters Acquity$^{TM}$ UPLC system equipped with a Waters Xevo$^{TM}$ G2 QTof MS (Milford, MA, USA). The detailed procedure was described in the supplementary methods.

The metabolic pathways that the differential metabolites were involved in were enriched using the Kyoto Encyclopedia of Genes and Genomes pathway (KEGG) tool, and significant altered pathways were represented by bubble chart using on-line bioinformatics tool (https://cloud.oebiotech.com/task).

**Fecal microbiota transplantation**. FMT was performed based on an established protocol[38] to evaluate the potential effect of microbiota, and termed as experiment 2. For this purpose, animals from experiment 1 was used as donor rats. Fecal contents were collected freshly and pooled from individual rats from sham, MN-vehicle, MN-HP (30 mg/kg) and MN-MMF groups at the end of the Experiment 1 24 h after the last dose of HP or MMF. 180−200 g recipient female SD rats were orally gavaged with donor fecal contents for first seven consecutive days and every two days for the remaining six weeks. Animals were randomly assigned to four different groups of six animals each: receipt-sham microbiota (R-Sham), receipt with MN-vehicle microbiota (R-vehicle), receipt with MN-MMF microbiota (R-MMF), receipt with MN-HP microbiota (R-HP).

The recipient rats were orally administrated with the supernatant of the fecal contents of the donor rats. After seven days of FMT, animals began to receive the intravenous c-BSA challenge. Aims to avoids prompt serious albuminuria induced by c-BSA, c-BSA injection was performed every two days for three weeks. The whole experiment last seven weeks, and at the end, the blood and urine were collected for biochemical analysis, and kidney and colon tissues were archived for pathology. Before sacrifice, the feces from each animal were collected for 16s sequencing. The detailed protocol was shown in Fig. 4a.

**In vitro metabolism studies**. For the LMs metabolism study, major coumarin derivates in HP, skimmin/apiosylskimmin (10 μM) and LMs (0.5 mg protein/mL) of human and rat were incubated together in Tris-HCl buffer (50 mM, pH 7.4). After the reaction was terminated, the residual skimmin/apiosylskimmin would be quantified by LC-MS/MS. Midazolam (10 μM) was used as the positive control.

To study the metabolism capability of gut microbiota, fresh rat feces were incubated anaerobically in the presence of Skimmin/ apiosylskimmin (10 μM). Similar as before, LC-MS/MS was performed to detect the remaining skimmin/apiosylskimmin.

**Effects on indole and p-cresol production by intestinal bacteria ex vivo**. Fresh cecal contents (1 g) were immediately suspended in 4 mL anaerobic medium. Then added the compound to be tested, with DMSO (8 μL) as the vehicle control. Test compounds and final concentrations were as follows: HP (1, 2, 4 mg/mL), 7-HC (4 mg/mL), skimmin (4 mg/mL), apiosylskimmin (4 mg/mL). Nitrophenyl-β-D-glucopyranoside (200 μM) was used as a positive control for intestinal metabolism. The cultures were incubated at 37 °C for 6 h in an anaerobic environment with $N_2$ atmosphere. All reactions were terminated with three volumes of ice-cold acetonitrile. LC-MS/MS was used to quantify the p-cresol and indole using Zorbax C18 column.

**Measurement of ROS production**. Rat mesangial cells HBZY-1 were pretreated with 1 and 10 μM 7-Hydroxycoumarin, skimmin and apiosylskimmin Then used the 2'−7'dichlorofluorescin diacetate (DCFH-DA, Sigma Co, St Louis, MO) to detect the ROS production inside the cell. ROS generation was determined by microplate reader (Tecan, Switzerland), with excitation wavelength of 485 nm and an emission wavelength of 530 nm.

**Western blot analysis**. Protein extraction from colon tissues was performed using a lysis buffer containing the inhibitor cocktail of phosphatase and protease according to the standard protocol. Same as mRNA isolation, aims to determine the gut barrier protein expression accurately, we used the sharp bladder to cut the inner layer down from intestinal segments, and to get more single-layer epithelial cells as much as possible. Primary

antibodies against claudin-1, mucin2, zonula occludens-1 (ZO-1), occludin and β-actin were purchased from abcam.

**Gut microbiota depletion.** To investigate whether HP played its renal protective effect dependent with gut microbiota, pseudo germ-free animals, the SD rats (160−180 g, female) were treated with an antibiotic cocktail (ABX) for the intestinal microbiota eradication[39]. In short, drinking water was supplied containing metronidazole (Macklin Biochemical, Shanghai, China), ampicillin (Macklin) and neomycin sulfate (Macklin), and vancomycin (Macklin) for six weeks. After three days of ABX treatment, the animals began to receive the c-BSA intravenous administration to induce the MN, and after proteinuria was established, HP 30 mg/ kg was given to animals in accordance with same procedure as Fig. S3. Meanwhile, animals receiving water were used as control group. The experimental details were presented in Fig.5a.

**Statistics and reproducibility.** Data was presented as mean ± standard deviation (SD). For the microbial diversity, QIIME (PAST 3×) was used to calculate the Shannon and Simpson indexes. For the multiple comparison, the variables were compared using one-way ANOVA and Tukey post hoc test for normal distribution, otherwise Mann–Whitney U test or Kruskal–Wallis with Dunn's multiple comparison test in case of abnormal distribution. $P < 0.05$ was considered significant. Prism (GraphPad Software) was used to conduct all statistical analyses. The specific statistical analysis performed, with all relevant information, is provided below the full dataset listed in the Supplementary Data 1 and supplementary data 2 file. Significance symbols (ranked as $P < 0.05$, $P < 0.01$, $P < 0.001$, $P < 0.0001$) are listed above the bars. All images shown are representative of at least 3 independent samples. All measurements shown in bar graphs were taken from distinct samples. No sample size calculation was performed. A minimum of $n = 3$ independent biological experiments were performed, with generally 6-10 performed per condition depending on the complexity/feasibility of the experiment. Blinding and randomization were not performed. For microscopy data, a minimum of 5 replicate fields per coverslip are imaged.

**Reporting summary.** Further information on research design is available in the Nature Portfolio Reporting Summary linked to this article.

## Data availability
Source data, as well as statistical analysis for all graphs, are provided in the Excel file Supplementary Data 1 and Supplementary Data 2. Source images for representative Western blots shown in figures are provided in Fig. 3 sheet and Fig. 4 sheet in Supplementary Data1.

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

## Acknowledgements

We appreciate the funding from Chinese academy of medical sciences Innovation Fund for Medical Sciences (CIFsMS, No. 2022-I2M-1-014), and National Natural Science Foundation of China (82293684), and National Key R&D Program of China (2022YFA0806400). We thank the Shanghai Luming biological technology co., LTD (Shanghai, China) for providing metabolomics services.

## Author contributions

Z.L. and H.W. carried out the animal experiments and mechanism study. L.S. performed pharmacokinetics study; J.M. and D.Z. was in charge of herbal preparation. X.L. and Z.M. was in charge of draft editing, animal study and manuscript submission. X.C. and S.Z. designed the whole study and drafted the manuscript. X.Z. carried out the metabolomics study, and was in charge of manuscript revision and illustration preparation. All data were generated in-house, and no paper mill was used. All authors agree to be accountable for all aspects of work ensuring integrity and accuracy.

## Competing interests

The authors declare no competing interests.

## Ethics approval

The animal experiments were approved by the Animal Care & Welfare Committee, Institute of Materia Medica, CAMS & PUMC (No. 002676).

## Consent for publication

All the authors have agreed the publication.
