## [Peer Review File · Communications Biology]

Hydrangea paniculata coumarins attenuate experimental membranous nephritis by bidirectional interaction with gut microbiotaReviewers' comments:

Reviewer #1 (Remarks to the Author):

The manuscript is generally very interesting and the main objective is clearly described. All sections of the manuscript were correctly addressed and in the information is relevant. There are only minor issues that need to be corrected in this manuscript.

1) The manuscript was mainly written in US English but it is still possible to find some words in UK English. Please correct.

2) There are some italics missing along the manuscript.

3) The graphical abstract and figure 8 should mention that this study is conducted with mice models, therefore the scheme is representative for such models.

4) Availability of data. The authors should make the raw data available for other researchers.

Therefore I suggest depositing the data on a public library data.

Reviewer #2 (Remarks to the Author):

The manuscript deals with the reciprocal interaction between the gut microbiota and coumarin-rich extract of *H. paniculate*. Although the authors have performed several experiments, the following points undermine the importance of the study. Therefore, the authors need to make substantial revisions as per the following comments.

- Avoid using the term 'dysbiosis'. The microbial phenotype related to 'eubiosis' is not properly defined. Therefore, it would be wise not to use the term dysbiosis
- The hypothesis behind the study is not strong enough. It is not clear what exactly prompted the authors to investigate the effects of the extract on gut microbiota?
- Line 640 and line 56: The authors have mentioned 'Hydrangea paniculate'. However, elsewhere, they have mentioned *Hydrangea paniculate*. Which one is correct?
- Why were only female animals used? How would the results be applicable to the human population that comprises both sexes.
- The rationale of dose selection and treatment duration is not mentioned clearly.
- The methodology for microbiome analysis needs clarification. What program was used to analyze the microbiome? What was the detailed pipeline? How were chimera sequences removed? What database was used? PCoA was based on which dataset?
- Fig 1C. The signs of histopathological injury should be specified using arrows. The appropriate methodological paper should be cited which was used to score the histopathology.
- Fig 3H. There are inconsistent data points for specific cohorts in each graph. Why?
- For gut microbial abundance data, how was 'zero' abundance score within individual cohorts treated in statistical analysis?
- The tight junction protein mRNA expressions were measured using whole intestinal segments. However, the single-layered epithelial cells make up the gut barrier. Therefore, how is the data from the whole intestinal segment relevant?
- Add Gen Bank accession number for all the primers

Reviewers' comments:

Reviewer #1 (Remarks to the Author):

The manuscript is generally very interesting and the main objective is clearly described. All sections of the manuscript were correctly addressed and in the information is relevant. There are only minor issues that need to be corrected in this manuscript.

1) The manuscript was mainly written in US English but it is still possible to find some words in UK English. Please correct.

Response: thanks, I have used nature editing language service to improve the whole manuscript writing.

2) There are some italics missing along the manuscript.

Response: thanks, I have read through the whole manuscript, and make some missing Latin words to be italics, such as *H.Paniculata*.

3) The graphical abstract and figure 8 should mention that this study is conducted with mice models, therefore the scheme is representative for such models.

Response: I have added the animal model description in the figure legend of figure 8. On the other hand, I also added this description into the figure.

4) Availability of data. The authors should make the raw data available for other researchers. Therefore I suggest depositing the data on a public library data.

Response: thanks, I have begun to prepare to deposit our data into the public database. because there are huge 16s rRNA sequencing data and metabolomics data, as well as pathological data in my study, it needs some time. I have checked the submission checklist of "communications biology", it is not mandatory before provisional acceptance. Once my manuscript has been accepted for publication, I will put the data deposit link in the final manuscript.

Reviewer #2 (Remarks to the Author):

The manuscript deals with the reciprocal interaction between the gut microbiota and coumarin-rich extract of *H. paniculate*. Although the authors have performed several experiments, the following points undermine the importance of the study. Therefore, the authors need to make substantial revisions as per the following comments.

- Avoid using the term 'dysbiosis'. The microbial phenotype related to 'eubiosis' is not properly defined. Therefore, it would be wise not to use the term dysbiosis

Response: thanks for your advice, but dysbiosis is often defined as an "imbalance" in the gut microbial community that is associated with disease. This imbalance could be due to the gain or loss of community members or changes in relative abundance of microbes. In my current study, the chronic kidney disease causes the remarkable changes of richness and diversity of intestinal microbes, therefore, we think the dysbiosis is suitable to describe this

condition. But if you think dysbiosis is not suitable, can you give me more suitable words? I am sorry that I do not know which word is more suitable to describe the imbalance of gut microbiota caused by CKD.

- The hypothesis behind the study is not strong enough. It is not clear what exactly prompted the authors to investigate the effects of the extract on gut microbiota?

Response: thanks for your advice, and actually we have a strong motivation to investigate the effects of the extract on gut microbiota. Based on our previous study (phytomedicine, 2022), the metabolite of HP extract, 7-hydroxycoumarin has more stronger bioactivity than prototype compounds. Later we confirm that 7-hydroxycoumarin was metabolized by gut microbiota but not liver, therefore, it suggests that HP renal protection effect may be partially dependent with gut microbiota metabolism. On the other hand, more and more data demonstrate that natural compounds can modulate the gut microbiota to play the beneficial effect, therefore, it motivates us to further investigate the bi-directional interaction between HP extract and gut microbiota.

- Line 640 and line 56: The authors have mentioned 'Hydrangea paniculate'. However, elsewhere, they have mentioned Hydrangea paniculata. Which one is correct?

Response: thanks for your advice, Hydrangea paniculata is correct, and there were two spelling errors in the full text, and I have corrected them in the revised manuscript.

- Why were only female animals used? How would the results be applicable to the human population that comprises both sexes.

Response: thanks for your advice, due to limited funding and time, we only chose one sex to perform the experiments. But I think the results are also applicable to the human population that comprised both sexes.

- The rationale of dose selection and treatment duration is not mentioned clearly.

Response: thanks for your advice, the extract of H. Paniculata has been developed as new herbal drug and move into pre-clinical stage currently. Therefore, the dose selection and treatment duration have been well determined based on our abundant preliminary studies and publications. I have added the rationale of dose selection and treatment duration into the revised manuscript, which you can find in Line, with highlight.

- The methodology for microbiome analysis needs clarification. What program was used to analyze the microbiome? What was the detailed pipeline? How were chimera sequences removed? What database was used? PCoA was based on which dataset?

Response: thanks. I am sorry that I did not describe them clear at the beginning. Actually, I have put the detailed methods into the supplemental documents, which you can download and read it. Now I have added some detailed methodology into the revised manuscript. KEGG pathway was analyzed by the annotation results of the amplicon which were correlated with the corresponding functional database, and PICRUST software was used for functional prediction and analysis of the microbial community in the ecological samples. Please find line 199 to line 225.

- Fig 1C. The signs of histopathological injury should be specified using arrows. The appropriate methodological paper should be cited which was used to score the histopathology.

Response: thanks, same answer as above. Actually, I have made a detailed description about

how to analyze the histopathology and cite the appropriate methodological publication, but the content was described in the supplemental documents. Then dear reviewer can download and read it. have put the several arrows to show the histopathological injury, such as protein cast, tubular vacuolation.

The following contents were added into the revised figure legend. "the blue circle refers to infiltration of inflammatory cells; Red arrows refers to protein cast and tubular dilatation; Black arrows refers to Masson dark staining with potential glomerulosclerosis; Yellow arrows refers to podocytes, which we can find there is almost no integral podocytes in MN model group treated by vehicle".

- Fig 3H. There are inconsistent data points for specific cohorts in each graph. Why?

Response: I am so sorry that I do not understand your advice well, for the figure 3H, I did not notice what is inconsistent data points for specific cohorts.

- For gut microbial abundance data, how was 'zero' abundance score within individual cohorts treated in statistical analysis?

Response: thanks, if a genus or species has some zero abundance score within individual, that means their abundance is quite low, and generally they are rare bacteria species. In this case, we did not put the bacteria with very low abundance into the analysis.

- The tight junction protein mRNA expressions were measured using whole intestinal segments. However, the single-layered epithelial cells make up the gut barrier. Therefore, how is the data from the whole intestinal segment relevant?

Response: thanks for your advice, and I am sorry to bring confusion for your understanding. Besides the intestinal segments, the kidney tissues were not described properly. For the mRNA and total protein isolation, we used the cortex of kidney, not the whole kidney, and I have added it in the revised manuscript. For the intestinal segment, actually, we use the sharp bladder to cut the inner layer down from intestinal segments, and to get more single-layer epithelial cells as much as possible, to determine the gut barrier protein expression accurately. I have added this into the revised method part of manuscript. Line 268-270, line 171-174.

- Add Gen Bank accession number for all the primers

Response: thanks, and I have added the gene bank accession number for all the primers into the revised supplemental data.

Reviewers' comments:

Reviewer #1 (Remarks to the Author):

In my opinion the manuscript has been improved and it can be published.

Reviewer #3 (Remarks to the Author):

Comments and Suggestions for Authors

In this manuscript, Zhaojun Li et al. evaluated the protective effects of *Hydrangea paniculate* on experimental membranous nephritis by fecal microbiota transplantation and pseudo germ-free animals, and stated bidirectional interaction between *Hydrangea paniculate* and gut microbiota contributes its renal beneficial effect. Nevertheless, the manuscript is confusing and poorly expressed, and requires substantial revision. I don't think this manuscript is suitable for publication in *Communications biology*.

Here are specific concerns:

Specific comments

1. The abbreviations in the manuscript were irregular, and the authors are advised to check and revise them entirely, including c-BSA and MMF. Just one occurrence does not require an abbreviation.
2. The logical coherence of the abstract is poor. The presentation of the previous studies of the group is insufficient, and the authors lack a comprehensive grasp of the need and significance of the study. In addition, certain concepts (e.g., bidirectional interactions of the gut microbiota) remain unarticulated.
3. Authors are responsible for checking the correctness of the citation of references, e.g., line 86, the original article seemed not to mention that the bioavailability of HP is 10%.
4. The methodology section exhibits irregularities in its writing, notably the absence of spaces between numbers and units, and the non-adherence to the International System of Units format.
5. It appears that the authors may have mistakenly used the term "bladder" instead of "blade." Also, is it feasible to obtain a single layer of epithelial cells using this method? The authors should consider providing additional literature references that support the potential success of this approach.
6. The scales of the histopathological sections in Figure 1 are poorly visualized and need to be revised.
7. In Figure 2, several expressions are too high and additionally there is no normalization of the data. It is recommended that authors recheck the related data.
8. The authors stated that HP has been reported that could alleviate membranous nephritis and affects gut microbiota, whereas the purpose of the present study is that HP exerts an ameliorative effect by affecting gut microbiota. It is advised that the authors reorganize the structure of the manuscript to make it clearer, e.g., by placing Figure 1 and its associated results in the supplementary material.
9. Certain English expressions are not suitable and need to be revised. It should be "In conclusion" instead of "In conclusions" in Line 45. A few expressions are unclear, unsuitable for a scientific essay, and tend to confuse the reader, including but not limited to Line 57-59.
10. Why did the author wrote "Total coumarins from *Hydrangea paniculate*" instead of "*Hydrangea paniculate*" in the title?

REVIEWERS' COMMENTS:

Reviewer #3 (Remarks to the Author):

Upon thorough review of the revised manuscript, it has been determined that despite the author's efforts to address previous concerns, the work still falls below the standards required for publication in *Communications Biology*. The manuscript's persistent fundamental flaws include a lack of clarity in methodology and experimental design, a noticeably superficial literature review, inadequate data analysis, and significant deficiencies in language and writing quality.

Reviewer #3 (Remarks to the Author):

Upon thorough review of the revised manuscript, it has been determined that despite the author's efforts to address previous concerns, the work still falls below the standards required for publication in Communications Biology. The manuscript's persistent fundamental flaws include a lack of clarity in methodology and experimental design, a noticeably superficial literature review, inadequate data analysis, and significant deficiencies in language and writing quality.

Response: thanks for your suggestions, and I have made a huge revision this time based on Nature's strict guidelines. And I make the graphs into dot-plot style, and make all the graphs have consistent format. On the other hand, I add more detailed description into the method part.

I also prepare all the checklist and all the original data as supplemental data set to upload.

I hope this time it can be accepted for publication.